# Clinical phenotypes and outcomes associated with SARS-CoV-2 variant Omicron in critically ill French patients with COVID-19

Infection with SARS-CoV-2 variant Omicron is considered to be less severe than infection with variant Delta, with rarer occurrence of severe disease requiring intensive care. Little information is available on comorbid factors, clinical conditions and specific viral mutational patterns associated with the severity of variant Omicron infection. In this multicenter prospective cohort study, patients consecutively admitted for severe COVID-19 in 20 intensive care units in France between December 7th 2021 and May 1st 2022 were included. Among 259 patients, we show that the clinical phenotype of patients infected with variant Omicron ($n = 148$) is different from that in those infected with variant Delta ($n = 111$). We observe no significant relationship between Delta and Omicron variant lineages/sublineages and 28-day mortality (adjusted odds ratio [95% confidence interval] = 0.68 [0.35–1.32]; $p = 0.253$). Among Omicron-infected patients, 43.2% are immunocompromised, most of whom have received two doses of vaccine or more (85.9%) but display a poor humoral response to vaccination. The mortality rate of immunocompromised patients infected with variant Omicron is significantly higher than that of non-immunocompromised patients (46.9% vs 26.2%; $p = 0.009$). In patients infected with variant Omicron, there is no association between specific sublineages (BA.1/BA.1.1 ($n = 109$) and BA.2 ($n = 21$)) or any viral genome polymorphisms/mutational profile and 28-day mortality.

The emergence of SARS-CoV-2 Variants of Concerns (VOCs) at the end of 2020 marked a turning point in the COVID-19 crisis that challenged and continues to challenge public health policies worldwide. The epidemiological situation has worsened with the emergence in November 2021 and subsequent rapid spread of the highly mutated Omicron variant and its sublineages. Although variant Omicron appears to cause a less severe disease than its predecessor variant Delta[1–4], with less frequent requirement for intensive care[1], a substantial number of Omicron-infected patients experienced severe COVID-19 with acute respiratory distress syndrome (ARDS) requiring intensive care support. A recent observational study has suggested that the mortality of critically ill patients infected with variant Omicron was not significantly different from that of patients infected with

variant Delta[5]. However, little is known on the comorbid factors and clinical conditions associated with the severity of variant Omicron infection. Specifically, the characteristics of patients with acute respiratory failure/ARDS admitted to intensive care units (ICUs) have not been reported and their impact on mortality remains unknown, making it difficult for health authorities to make informed decisions.

Variant Omicron expansion has been characterized by the emergence and subsequent spread of several sublineages (initially BA.1, BA.1.1, and BA.2; more recently BA.4 and BA.5). Compared to previous SARS-CoV-2 strains, Omicron variants carry a large number (26 to 32) of non-synonymous mutations (substitutions and/or deletions) in their spike (S) protein gene, and approximately 20 additional mutations in other structural and non-structural genes[6,7]. Although several of these

✉ e-mail: slim.fourati@aphp.fr

mutations have already been observed in previous variants of concern, the majority of Omicron mutations are unique and their impact of disease severity is unknown.

In the present study, we compared the characteristics of critically ill patients with acute respiratory failure infected with variant Omicron with those of patients infected with variant Delta and explored the relationship between Omicron sublineages and specific viral mutations/mutational patterns with the clinical features and outcomes of COVID-19 disease in these patients.

## Results

Between December 7th, 2021, and May 1st, 2022, 377 patients were admitted in one of the 20 participating ICUs and included in the study. Variant lineage was identified in 310 of them, while day-28 follow-up was available in 259, including 111 patients infected with variant Delta and 148 patients with variant Omicron. Full-length viral genome sequence analysis yielding high coverage (>90%) was performed in 97 of the 148 Omicron-infected patients. Figure 1 illustrates the dynamics of emerging SARS-CoV-2 lineages and sublineages during the study period. Variant Delta was predominant in December 2021 (i.e., during the first 5 weeks of the inclusion period), but was then rapidly replaced by variant Omicron, sublineages BA.1 and BA.1.1 which became predominant in the second week of January 2022, then by lineage BA.2, which became dominant in March 2022 and remained so until the end of the inclusion period.

### Clinical phenotype and vaccination background of Omicron vs Delta-infected patients

Compared to patients infected with variant Delta, those infected with variant Omicron were older and had significantly more frequent comorbidities, including hypertension, chronic respiratory failure, and chronic renal failure, together with a higher clinical frailty scale (Table 1). Three times more Omicron- than Delta-infected patients were immunosuppressed (43.2% ($n = 64/148$) vs 13.6% ($n = 15/111$), respectively; $p < 0.0001$).

Patients infected with variant Omicron had significantly more often been vaccinated than those infected with variant Delta (at least one dose received: 60.1% ($n = 89/148$) vs 29.1% ($n = 32/111$),

respectively; $p < 0.0001$) (Table 1). Vaccinated Omicron-infected patients had received a significantly higher number of vaccine doses (half of them had received 3 doses of vaccine) than Delta-infected patients (Table 1). However, the proportion of patients with detectable anti-S antibodies at ICU admission was not significantly different between the groups, possibly because of the high proportion of immunosuppressed patients in the better vaccinated Omicron group.

There was a marked difference between the Omicron and Delta groups regarding the severity of the disease at ICU admission. Indeed, patients infected with variant Omicron exhibited higher SOFA and SAPS II scores, as a consequence of more severe extra-pulmonary organ failures, including renal and hematological failures (Fig. 2). In contrast, although the difference in need for invasive mechanical ventilation support was not significant between the two groups, patients infected with variant Delta had more severe respiratory failure at ICU admission, with more ARDS with refractory hypoxemia requiring veno-venous extracorporeal membrane oxygenation (ECMO) (8.2% ($n = 9/111$) for Delta vs 1.4% ($n = 2/148$) for Omicron, respectively; $p = 0.011$) (Table 1).

To better characterize the phenotypic differences between patients infected with variants Omicron and Delta, we used the SOM methodology to plot 2-D maps of patients grouped according to their clinical and biological characteristics at ICU admission (Fig. 3). SOM analysis confirmed the differences between Delta- and Omicron-infected patients. As shown in the figure, patients infected with variant Omicron clustered in the upper left area of the map, where the highest frequencies of immunosuppression and vaccination also clustered. Patients infected with variant Delta were distinctly located in the lower half area of the map. Patients with the highest rates of day-28 mortality clustered in the upper right area of the map, where the highest values of the SOFA and SAPS II scores also clustered together with a high frequency of comorbidities and immunosuppression, involving a mixture of both Delta- and Omicron-infected patients.

### Influence of infecting variants (Delta vs Omicron) on management and day-28 mortality

Half of the patients (51.0%, $n = 132/259$) required invasive mechanical ventilation during their ICU stay, with no significant difference

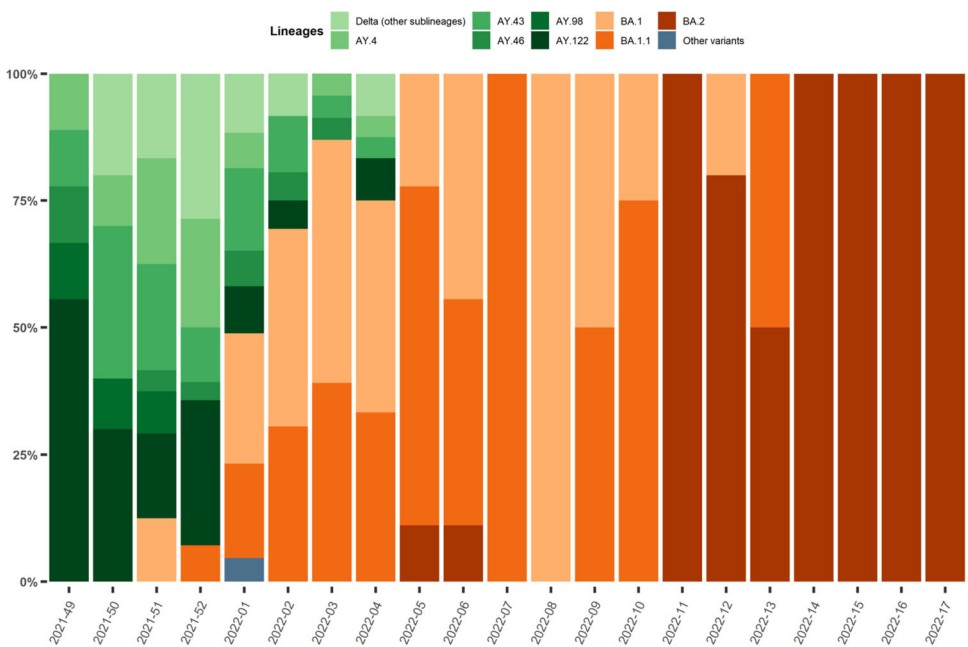

**Fig. 1 | Dynamics of infecting SARS-CoV-2 variants during the study period in patients requiring intensive care hospitalized in the 20 participating centers.** Delta lineages/sublineages are in green, Omicron lineages/sublineages are in orange/red. VOC: variants of concern; Source data are provided as a Source Data file.

**Table 1 | Clinical and biological characteristics of the 259 patients with severe SARS-CoV-2 infection at the time of their intensive care unit admission according to the infecting SARS-CoV-2 variant (Delta vs Omicron)**

| | | All patients N = 259 | Delta N = 111 | Omicron N = 148 | p-value |
|---|---|---|---|---|---|
| **Demographics and comorbidities** | | | | | |
| Sex, females | | 82 (31.66%) | 40 (36.04%) | 42 (28.38%) | 0.190 |
| Age, years | | 61.2 (±12.8) | 57.6 (±14.5) | 63.9 (±10.8) | **<0.0001** |
| Diabetes | | 86 (33.46%) | 35 (31.82%) | 51 (34.69%) | 0.629 |
| Obesity | | 99 (38.52%) | 48 (43.24%) | 51 (34.93%) | 0.175 |
| Chronic heart failure | | 21 (8.14%) | 6 (5.45%) | 15 (10.14%) | 0.174 |
| Hypertension | | 118 (45.74%) | 42 (38.18%) | 76 (51.35%) | **0.036** |
| Chronic respiratory failure | | 23 (8.91%) | 4 (3.64%) | 19 (12.84%) | **0.010** |
| Chronic renal failure | | 46 (17.97%) | 12 (10.91%) | 34 (23.29%) | **0.011** |
| Cirrhosis | | 4 (1.55%) | 1 (0.91%) | 3 (2.03%) | 0.472 |
| Immunosuppression (3 cat.) | None | 179 (69.38%) | 95 (86.36%) | 84 (56.76%) | **<0.0001** |
| | Solid organ transplant | 40 (15.50%) | 8 (7.27%) | 32 (21.62%) | |
| | Onco-hematological malignancies | 21 (8.14%) | 2 (1.82%) | 19 (12.84%) | |
| | Others[a] | 18 (6.98%) | 5 (4.55%) | 13 (8.78%) | |
| Number of comorbidities | | 2 (1;3) | 1 (0;2) | 2 (1;3) | **<0.0001** |
| Clinical frailty scale | | 3 (2;4) | 3 (2;3) | 3 (3;4) | **0.005** |
| **SARS-CoV-2 infection and Vaccination** | | | | | |
| Previous SARS-CoV-2 infection | | 16 (6.23%) | 10 (9.09%) | 6 (4.08%) | 0.100 |
| SARS-CoV-2 vaccination | | 121 (46.90%) | 32 (29.09%) | 89 (60.14%) | **<0.0001** |
| Number of doses among vaccinated | | 3 (2;3) | 2 (2;3) | 3 (2;3) | **0.005** |
| Last dose - ICU admission[b], days | | 135 (39;210) | 73 (24;174) | 151.50 (57;217) | **0.038** |
| SARS-CoV-2 serology at ICU admission | Unavailable | 96 (37.07%) | 35 (31.53%) | 61 (41.22%) | 0.275 |
| | Negative[c] | 91 (35.14%) | 43 (38.74%) | 48 (32.43%) | |
| | Positive | 72 (27.80%) | 33 (29.73%) | 39 (26.35%) | |
| First symptoms - ICU admission, days | | 8 (6;11) | 9 (7;11) | 7 (4.50;10) | **0.006** |
| SARS-CoV-2 RNA detection in nasopharyngeal swabs, Ct | | 22 (19;26) | 22 (20;26) | 22 (19;27) | 0.643 |
| **Patients severity upon ICU admission and biological features** | | | | | |
| WHO 10-point scale | | 6 (6;6) | 6 (6;7) | 6 (6;6) | 0.467 |
| SAPS II score | | 34 (26;43) | 31 (23;40) | 35 (28;45) | **0.022** |
| SOFA score | | 4 (3;6) | 4 (2;5) | 4 (3;6) | **0.014** |
| $PaO_2/FiO_2$ ratio, mmHg | | 108 (74;167) | 106 (69;178) | 109 (78;158) | 0.781 |
| Arterial lactate level, mM | | 1.60 (1.10;2.20) | 1.60 (1.10;2.20) | 1.60 (1.10;2.15) | 0.676 |
| Blood leukocytes, G/L | | 8.20 (5.10;12.40) | 9.00 (5.70;12.50) | 7.80 (4.85;12.00) | 0.187 |
| Blood lymphocytes, G/L | | 0.60 (0.40;0.90) | 0.70 (0.40;1.10) | 0.50 (0.30;0.80) | **0.001** |
| Blood platelets, G/L | | 221 (155;301) | 248 (200;332) | 193 (133;278) | **<0.0001** |
| Serum urea level, mM | | 8 (5;15) | 7 (5;12) | 9 (6;16) | **0.007** |
| Serum creatinine level, µM | | 83 (63;129) | 73 (59;106) | 95 (67;159) | **<0.0001** |
| Bacterial coinfection | | 36 (13.90%) | 15 (13.51%) | 21 (14.19%) | 1.000 |
| Pulmonary embolism | | 16 (6.32%) | 7 (6.54%) | 9 (6.16%) | 1.000 |
| Lung parenchyma involvement, % | | 50 (40;75) | 50 (50;75) | 50 (40;75) | 0.880 |
| Oxygen/ventilatory support | Oxygen | 44 (16.99%) | 17 (15.32%) | 27 (18.24%) | 0.553 |
| | High flow oxygen | 133 (51.35%) | 57 (51.35%) | 76 (51.35%) | |
| | NIV/C-PAP | 21 (8.11%) | 7 (6.31%) | 14 (9.46%) | |
| | Invasive MV | 61 (23.55%) | 30 (27.03%) | 31 (20.95%) | |
| ECMO | | 11 (4.28%) | 9 (8.18%) | 2 (1.36%) | **0.008** |
| Vasopressor support | | 33 (13.15%) | 14 (12.84%) | 19 (13.38%) | 0.901 |

Results are N (%), means (±standard deviation) or medians (interquartile range, i.e., quartile 1; quartile 3).
ICU intensive care unit, Ct cycle threshold, WHO World Health Organization, SOFA Sequential Organ Failure Assessment, SAPS II Simplified Acute Physiology Score II, NIV non-invasive ventilation, C-PAP continuous-positive airway pressure, MV mechanical ventilation, ECMO extracorporeal mechanical ventilation.
[a]Includes HIV infection, long-term corticosteroid treatment, and other immunosuppressive treatments.
[b]Time lag between the last vaccination dose and ICU admission.
[c]Defined as <30 Binding Antibody Units (BAU)/mL.
Two-tailed p-values come from unadjusted comparisons using Chi-square or Fisher's exact tests for categorical variables, and t-tests or Mann–Whitney tests for continuous variables, as appropriate.
No adjustment for multiple comparisons was performed; Bolded p-values are significant at the $p < 0.05$ level.

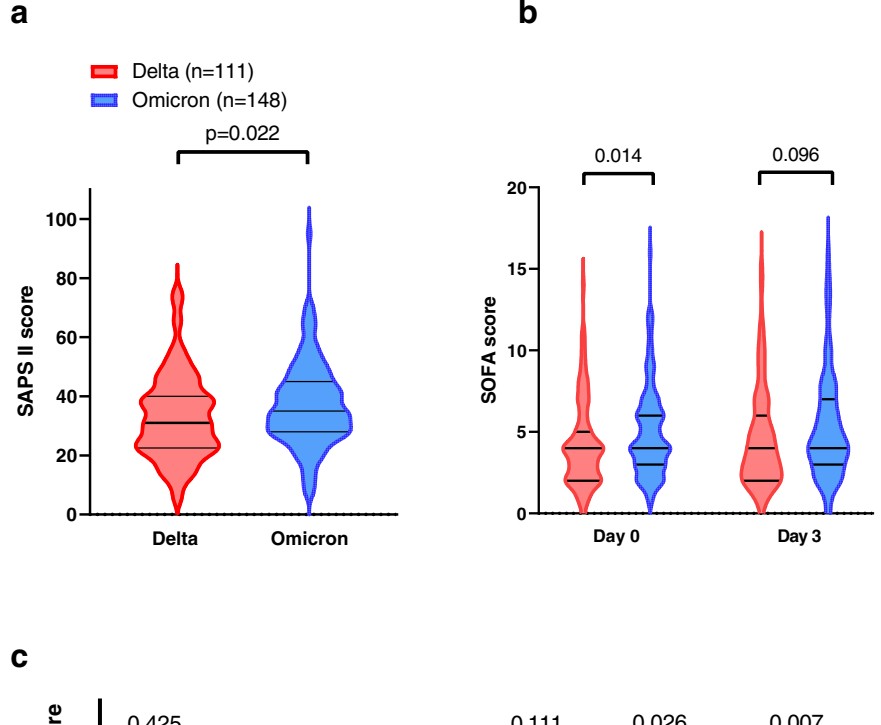

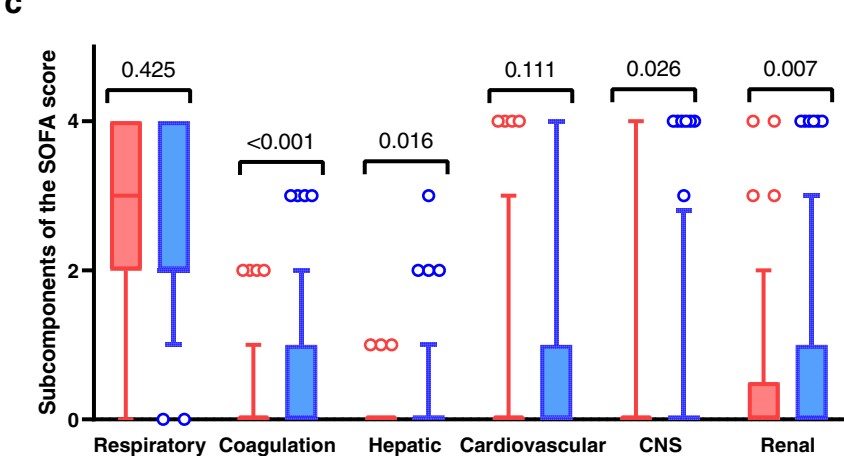

**Fig. 2 | Severity of illness scores in patients infected with variant Delta (red; n = 111) and variant Omicron (blue; n = 148). a** SAPS II scores at ICU admission; **b** SOFA score at day 0 (admission at the ICU) and day 3 of hospitalization (by two-way ANOVA, there was a significant effect of the variant (p = 0.0269), no significant effect of time (p = 0.2784), and no significant interaction of both parameters (variant x time; p = 0.7668)). In (**a**) and (**b**), data distribution is represented using violin plots and horizontal bars show the median and quartiles 25 and 75; **c** Organ system components of the SOFA score at ICU admission. Data distribution is represented using box-and-whisker plots, displaying median values and quartiles 25 and 75 (i.e., lower and upper limits of the box), and 5 and 95% percentiles (circles). Two-sided p-values have been generated with the Mann–Whitney test or the Sidak post-hoc ANOVA test; n = 111 and 148 independent measurements in the Delta and Omicron groups, respectively. SOFA: Sequential Organ Failure Assessment; SAPS II: Simplified Acute Physiology Score II; Source data are provided as a Source Data file.

between patients infected with variants Omicron or Delta (Table 2). Nevertheless, Delta-infected patients more often needed veno-venous ECMO support (18.0% (n = 20/111) vs 6.1% (n = 9/148) in Omicron-infected patients, respectively; p = 0.003). As far as COVID-19 specific management was concerned, there was no significant difference regarding dexamethasone administration, but patients in the Omicron group received significantly less tocilizumab than those in the Delta group (35.2% (n = 49/148) vs 55.8% (n = 58/111), respectively; p = 0.001). Monoclonal antibodies were similarly used in both groups of patients (19.9% of patients overall; n = 48/259). Patients infected with variant Delta almost exclusively received casirivimab–imdevimab (80.0%, n = 16/20), while those infected with variant Omicron mostly received tixagevimab–cilgavimab (85.7%, n = 24/28).

There was no difference in the frequency of day-28 mortality between Omicron- and Delta-infected patients (35.1% (n = 52/148) vs 28.8% (n = 32/111); p = 0.283) (Table 2). Univariable (Supplementary

Table 1) and multivariable (Table 3) logistic regression analyses showed that the variant lineage (Omicron vs Delta) was not significantly associated with day-28 mortality, whereas age and SOFA score were.

**Relationship between SARS-CoV-2 mutations/deletions and day-28 mortality in patients infected with variant Omicron**
Over the full-length viral genomes of 97 Omicron sample-related patients, 199 non-synonymous mutations (leading to amino acid substitutions) or deletions were detected in at least one sample. We first focused on mutations previously reported to correlate with worse clinical outcomes (P681R in Spike, P25L and Q57H in Orf3a and S194L, R203K or G204R in N)[8–11]. We found that none of these substitutions correlated with day-28 mortality. We then investigated non-biased amino acid modifications (substitutions or deletions) in Spike and outside Spike protein found in at least 20 viral genome of patients infected by Omicron (n = 11 positions) and analyzed the

## Patients characteristics at ICU admission

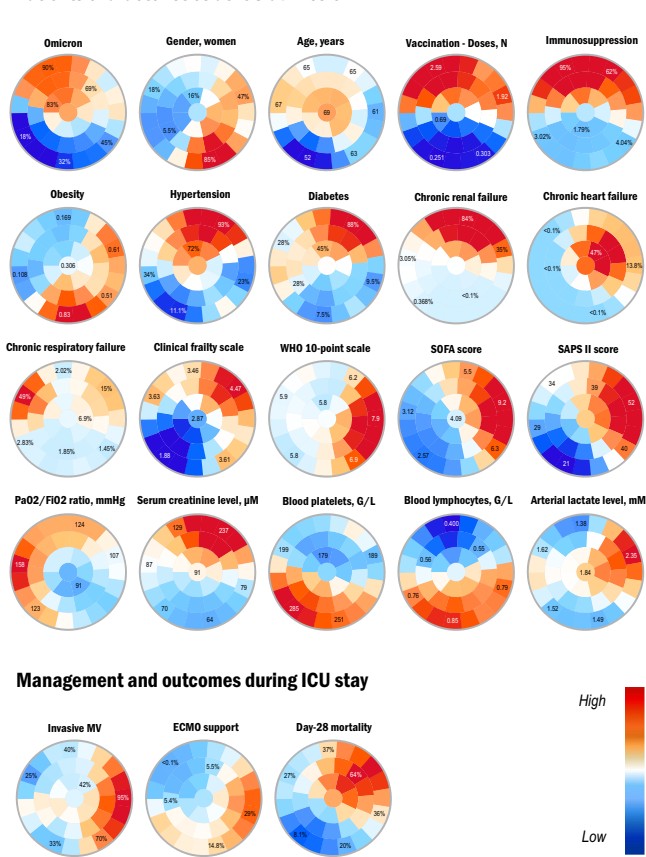

## Management and outcomes during ICU stay

**Fig. 3 | Unsupervised analysis of the clinical and biological characteristics of the patients infected with variants Omicron and Delta by self-organized maps (SOMs).** Unsupervised analysis by SOM automatically located patients with similar clinical and paraclinical parameters within 1 of 40 small groupings ("districts") throughout the map. The more similar the patients, the closer on the map. Each individual map shows the mean values or proportions per district for each characteristic: blue indicates the lowest average values, red the highest, with numbers shown for a selection of representative districts in each SOM. In the upper left area of the maps were located mostly Omicron-infected patients, as shown by the high prevalence rates (red colors) in this area of the Omicron map, whereas patients infected with variant Delta were distinctly located in the lower half area, as indicated by low Omicron prevalence rates (blue colors) in this area. Interpretation of the key characteristics associated with Omicron versus Delta patients can be drawn from the observation of the other maps: for instance, Omicron-infected patients in the upper left area also had higher immunosuppression rates and mean number of vaccination doses received as indicated by the red colors in the corresponding maps. WHO World Health Organization, SOFA Sequential Organ Failure Assessment, SAPS II Simplified Acute Physiology Score II, MV mechanical ventilation, Source data are provided as a Source Data file.

relationship between these mutations and day-28 mortality in univariable analysis. Two amino acid substitutions (ORF1a-K856R = NSP3 K38R and M-D3G) were associated with day-28 mortality but this relationship was no longer statistically significant after correction of p-values for test multiplicity using the Benjamini–Hochberg procedure (Fig. 4).

### Influence of immunosuppression on the severity of the disease in patients infected with variant Omicron

Overall, 43.2% of patients infected with variant Omicron had an underlying immunosuppression. Half of them (50.0%, n = 32/64) had received an organ transplant, while 29.7% (n = 19/64) had an onco-hematological malignancy (Table 1).

Compared to non-immunocompromised patients, immunocompromised ones had more frequent comorbidities, higher clinical frailty scales, higher severity of illness scores, and higher SARS-CoV-2 RNA levels (reflected by lower nasopharyngeal RT-PCR Ct values) at ICU admission (Table 4). Immunosuppressed patients had been more frequently vaccinated than their non-immunocompromised counterparts (85.9% (n = 55/64) vs 40.5% (n = 34/84), respectively; p < 0.0001). However, there was no statistically significant difference between the two groups regarding the proportion of patients with detectable anti-SARS-CoV-2 spike (S) antibodies at ICU admission. The serum titer of anti-S antibodies of vaccinated vs non-vaccinated immunocompromised patients was also not significantly different (mean difference [95% CI]: 1078 Binding Antibody Units (BAU)/mL [−319.4;2475.0]; p = 0.160), suggesting poor antibody response to vaccination in this population, whereas it was significantly different between vaccinated and non-vaccinated immunocompetent individuals (mean difference [95% CI]: 1062 BAU/mL [365.0;1760.0]; p = 0.0015) (Fig. 5), in keeping with deficient antibody response to vaccination in immunosuppressed patients. Importantly, the median time elapsed between the last vaccine dose and ICU admission did not significantly differ between immunocompromised and non-immunocompromised patients (175 [86–220] vs 116 [39–210] days, respectively; p = 0.191).

Among patients infected with the Omicron variant, those who were immunocompromised developed more frequent organ failures than those who were not during the course of their ICU stay (63.6% vs 45.5%, respectively; p = 0.025), while the rate of day-28 mortality was significantly higher (46.9% (n = 30/64) vs 26.2% (n = 22/84), respectively; p = 0.009) (Table 5, Supplementary Table 2). However, in multivariable analysis, the only variables significantly associated with day-28 mortality were age, SOFA score, chronic heart failure and diabetes (Supplementary Table 3).

Immunocompromised patients were less likely to receive anti-interleukin (IL)-6 receptor antagonists, whereas they more frequently received monoclonal antibodies (Table 5). An exploratory analysis assessing the effect of monoclonal antibodies in immunocompromised patients infected with variant Omicron did not reveal a significant association with 28-day mortality (52.6% (n = 10/19) vs 45.2% (n = 19/42) in those treated and not treated, respectively; p = 0.592).

### Lack of protection by SARS-CoV-2 vaccination against critical illness in immunocompromised patients infected with variant Omicron BA.2

Omicron BA.1 and BA.1.1 sublineages co-existed during the first 10 weeks of 2022, and were then both replaced by lineage BA.2 (Fig. 1). Patients infected with BA.2 more frequently had prior chronic respiratory failure than patients infected with other variants (38.1% (n = 8/21) vs 5.5% (n = 6/109), respectively; p = 0.0002). They were immunosuppressed in 57.1% of cases and had more frequent bacterial co-infections than other patients at ICU admission (28.6% (n = 6/21) vs 11.9% (n = 13/109), respectively; p = 0.048).

Patients infected with BA.2 also had a very high anti-SARS-CoV-2 vaccination coverage (85.7%), with a median number of doses received of 3 (IQR, 3–3) (Supplementary Table 4). Although there was no statistically significant difference in day-28 mortality between the different Omicron sublineages, patients infected with BA.2 had a numerically lower mortality rate (19.0%) than patients infected with other sublineages.

## Discussion

This prospective multicenter study included 307 critically ill patients with COVID-19 hospitalized in 20 French ICUs during winter and early spring 2022, a period during which variant Delta was progressively replaced by different lineages/sublineages of variant Omicron (BA.1, BA.1.1 and BA.2). The main results of our study are the following: (i) Patients infected with variant Omicron were characterized by a

**Table 2 | Intensive care management and outcomes of patients with severe SARS-CoV-2 infection (*n* = 259) during their intensive care unit stay according to the SARS-CoV-2 infecting variant (Delta or Omicron)**

| | | All patients N = 259 | Delta N = 111 | Omicron N = 148 | *p*-value |
|---|---|---|---|---|---|
| Invasive MV | | 132 (50.97%) | 63 (56.76%) | 69 (46.62%) | 0.106 |
| Prone positioning | | 101 (41.74%) | 51 (48.11%) | 50 (36.76%) | 0.076 |
| MV duration, days | | 16 (6;29) | 19 (8;34) | 12.50 (5;20) | **0.019** |
| ECMO support | | 29 (11.20%) | 20 (18.02%) | 9 (6.08%) | **0.003** |
| Duration of ECMO, days | | 27 (10;48) | 35.50 (12.50;63.50) | 8 (8;25) | **0.047** |
| Vasopressor support | | 113 (43.97%) | 54 (48.65%) | 59 (40.41%) | 0.188 |
| Duration of vasopressors, days | | 5.50 (2;16) | 8.50 (3.50;27.50) | 4 (1;12) | **0.016** |
| Renal Replacement Therapy | | 39 (15.06%) | 14 (12.61%) | 25 (16.89%) | 0.341 |
| Ventilator-acquired pneumonia (among IMV) | | 86 (67.19%) | 45 (73.77%) | 41 (61.19%) | 0.130 |
| Number of VAP episodes | Median (IQR) | 1 (0;2) | 1 (0;2) | 1 (0;2) | 0.100 |
| | 0 | 42 (33.07%) | 16 (26.67%) | 26 (38.81%) | **0.020** |
| | 1 | 42 (33.07%) | 22 (36.67%) | 20 (29.85%) | |
| | 2 | 28 (22.05%) | 10 (16.67%) | 18 (26.87%) | |
| | 3 | 15 (11.81%) | 12 (20.00%) | 3 (4.48%) | |
| CAPA | | 18 (7.06%) | 11 (10.09%) | 7 (4.79%) | 0.102 |
| Dexamethasone | | 214 (88.43%) | 93 (90.29%) | 121 (87.05%) | 0.436 |
| Tocilizumab | | 107 (44.03%) | 58 (55.77%) | 49 (35.25%) | **0.001** |
| Monoclonal antibodies | | 48 (19.92%) | 20 (19.23%) | 28 (20.44%) | 0.816 |
| Casirivimab–Imdevimab | | 16 (33.33%) | 16 (80.00%) | 0 (0.00%) | **<0.0001** |
| Tixagevimab–Cilgavimab | | 26 (54.17%) | 2 (10.00%) | 24 (85.71%) | **<0.0001** |
| Sotrovimab | | 4 (8.33%) | 0 (0.00%) | 4 (14.29%) | 0.077 |
| Day-28 mortality | | 84 (32.43%) | 32 (28.83%) | 52 (35.14%) | 0.283 |
| Duration of ICU stay, days | | 11 (6;22) | 11 (6;27.50) | 11 (5;20) | 0.132 |

Results are N (%), means (±standard deviation) or medians (interquartile range, i.e., quartile 1; quartile 3).
*MV* mechanical ventilation, *ECMO* extracorporeal mechanical ventilation, *VAP* ventilator-acquired pneumonia, *IMV* invasive mechanical ventilation, *CAPA* COVID-19-associated pulmonary aspergillosis,
[a]VAP episodes were recorded per definition in patients under IMV since more than 48 h.
Two-tailed *p*-values come from unadjusted comparisons using Chi-square or Fisher's exact tests for categorical variables, and *t*-tests or Mann–Whitney tests for continuous variables, as appropriate.
No adjustment for multiple comparisons was performed; Bolded *p*-values are significant at the *p* < 0.05 level.

**Table 3 | Independent predictors of day-28 mortality available within 24 h after ICU admission by multivariable logistic regression analysis in the 259 patients with Omicron or Delta infection**

| | | aOR (CI95%) | *p*-value |
|---|---|---|---|
| Age, years | | 1.06 (1.03;1.09) | **<0.0001** |
| SOFA score | | 1.20 (1.07;1.34) | **0.002** |
| Chronic heart failure | | 4.47 (1.50;13.34) | **0.007** |
| Sex, females | | 0.83 (0.43;1.62) | 0.585 |
| Immunosuppression | | 1.60 (0.76;3.35) | 0.213 |
| Variant | Delta | 1(ref) | |
| | Omicron | 0.68 (0.35;1.32) | 0.253 |
| SARS-CoV-2 vaccination | | 1.49 (0.73;3.04) | 0.275 |

aOR (CI 95%): adjusted Odds Ratio (95% confidence interval).
*SOFA* Sequential Organ Failure Assessment.
*p*-values come from multivariable logistic regression models; Bolded *p*-values are significant at the *p* < 0.05 level.

different clinical phenotype when compared to those infected with variant Delta, and more of them had been vaccinated; (ii) There was no statistically significant association between main variant lineages and 28-day mortality; (iii) Forty-three percent of patients infected with variant Omicron were immunocompromised, and their mortality rate was almost twice as high as that of non-immunocompromised patients; (iv) In patients infected with variant Omicron, there was no

association between specific sublineages (BA.1, BA.1.1 and BA.2) or any viral genome polymorphism or mutational profile and 28-day mortality.

Critically ill patients infected with variant Omicron displayed striking phenotype differences with those infected with variant Delta. Indeed, they were older, frailer with more comorbidities, including immunosuppression, and had higher severity of illness scores at ICU admission, reflecting more extra-pulmonary organ failures. Patients infected with variant Omicron had a different vaccination background from those infected with Delta, the vaccinated former having been more frequently vaccinated and those vaccinated having received more doses.

Our study is, to the best of our knowledge, the first one to accurately and extensively describe the clinical presentation of critically ill patients with COVID-19 infected with variant Omicron. Bouzid et al. compared the clinical presentations of patients presenting to the emergency department with Delta and Omicron variant infections. They did not observe the same differences between groups[1], suggesting that the sub-population of ICU patients exhibits specific clinical features.

We found no association between viral characteristics, including the main variant lineages, and mortality. This result contrasted with previous findings in the general population. Indeed, infection with variant Omicron was shown in recent observational cohort studies to be associated with a reduced risk of hospitalization[1,2,12,13] and death[1,13,14], as compared to Delta infection. Our study is in keeping with a recent one[5], indicating that there is no significant difference between variants Omicron and Delta in terms of mortality when focusing on the specific

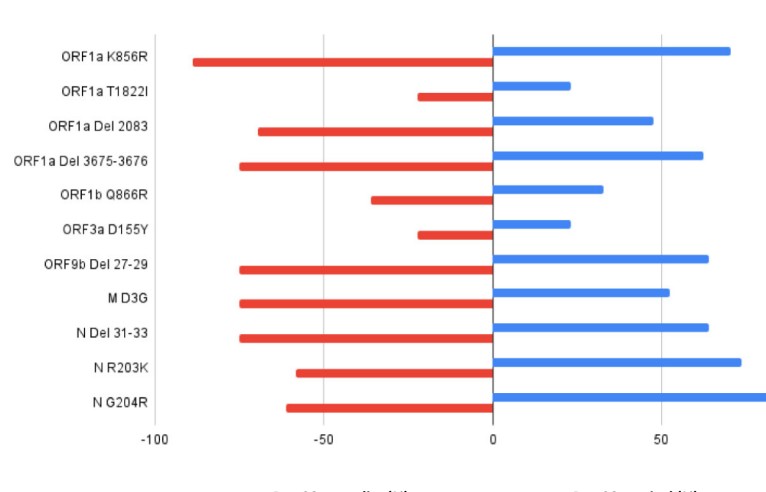

| Presence of mutation (Total N= 97) | OR (CI95%) | Raw p-value | Corrected p-value* |
|---|---|---|---|
| 75 | **3.35 (1.03;10.86)** | **0,044** | 0,154 |
| 22 | 0.96 (0.36;2.57) | 0,934 | 0,934 |
| 54 | 2.51 (1.05;5.98) | 0,334 | 0,492 |
| 65 | 1.82 (0.73;4.53) | 0,201 | 0,485 |
| 33 | 1.16 (0.49;2.75) | 0,739 | 0,776 |
| 22 | 0.96 (0.36;2.57) | 0,934 | 0,934 |
| 66 | 1.69 (0.68;4.24) | 0,261 | 0,485 |
| 59 | **2.72 (1.10;6.73)** | **0,031** | 0,154 |
| 66 | 1.69 (0.68;4.24) | 0,261 | 0,485 |
| 66 | 0.50 (0.21;1.19) | 0,118 | 0,381 |
| 72 | **0.35 (0.14;0.88)** | **0,026** | 0,154 |

**Fig. 4 | Relationship between SARS-CoV-2 mutations/deletions and day-28 mortality in patients infected with variant Omicron.** Over the full-length viral genomes of 97 Omicron sample-related patients, we investigated amino acid modifications (substitutions or deletions) outside Spike protein found in at least 20 viral genomes of patients infected by Omicron (*n* = 11 positions) and analyzed the relationship between these mutations and day-28 mortality in univariable analysis. Two amino acid substitutions (ORF1a-K856R = NSP3 K38R and M-D3G) were significantly associated with day-28 mortality but this relationship was no longer significant after correction of *p*-values for test multiplicity using the Benjamini–Hochberg procedure. Day 28 mortality is displayed in red and day 28 survival in blue; *p*-values come from unadjusted logistic regression modeling. Horizontal bars represent percentages; Source data are provided as a Source Data file.

population of patients with severe COVID-19 pneumonia requiring ICU admission, despite the fact that this condition is rarer in patients with Omicron infection.

In the Omicron group, immunocompromised patients represented up to 43% of cases, and their mortality rate was high (47%). We showed that vaccinated immunocompromised patients had a poor humoral response, with no significantly different serum anti-S titers following vaccination (two or three doses) from non-vaccinated patients. Whether additional booster vaccine doses would enhance the humoral response in these patients remains unknown. Monoclonal antibodies were used in one-third of immunocompromised patients, following the positive results of the RECOVERY trial[15]. However, no differences in outcomes were observed with or without treatment, suggesting that therapy could have been initiated too late during the course of the infection. Overall, our results point to the need for targeting COVID-19 prevention measures to the high-risk of critical illness and death immunocompromised population. These measures include prevention of infection, reinforcement of early preemptive treatments with antiviral agents (e.g., paxlovid and/or other direct-acting antiviral agents that were not used in our study), and early administration of variant-active monoclonal antibodies.

Here, we report for the first time the clinical phenotypes associated with different Omicron sublineages, in particular in the yet never described group of patients infected with variant Omicron BA.2 requiring ICU admission. These patients had been fully vaccinated in 86% of cases. However, they often had chronic respiratory failure and were more often immunocompromised. Noticeably, the mortality rate in patients infected with Omicron BA.2 was numerically lower than that observed for other Omicron lineages. Because BA.4 and BA.5 sublineages derive from BA.2[16] and these variants have become dominant after the end of the inclusion period of this study[17], our results are reassuring if confirmed that the effect of these variants on the clinical severity of the disease is similar to that of BA.2. Overall, the clinical phenotype of BA.2-infected patients reported here further emphasizes the crucial role of implementing early preventive therapy in immunocompromised patients to prevent severe cases. We found no statistically sound association between SARS-CoV-2 mutations

(deletions/ substitutions) in Omicron sublineages and day-28 mortality, highlighting the role of host factors rather than virus variability in the pathophysiology of severe diseases.

Our study has some limitations. The relatively small number of BA.2-infected patients included may have limited our statistical ability to show between-group differences. In-depth mutation analysis could not be performed for all patients, because full-length viral genome sequences were analyzed only when the sequencing coverage was greater than 90%. However, our study also has strengths, including the constitution of a unique prospective multicenter cohort of well-phenotyped critically ill patients and the availability of full-length SARS-CoV-2 genome sequences generated by up-to-date technology.

In conclusion, critically ill patients infected with variant Omicron exhibited a different clinical phenotype from that observed in patients infected with variant Delta. However, the variant lineage had no effect on day-28 mortality in this population. Immunocompromised patients infected with variant Omicron, in particular its BA.2 sublineage, represented a high-risk group with a poor humoral response to vaccination. In patients infected with variant Omicron, we found no association between specific sublineages (BA.1, BA.1.1 and BA.2) or any viral genome polymorphism or mutational profile and day-28 mortality.

## Methods
### Study design and patients
This is a prospective multicenter observational cohort study. Patients admitted between December 7th, 2021 (week 49/21) and May 1st, 2022 (week 17/22) in one of the 20 participating ICUs (17 from the Greater Paris area and 3 from the North-East of France) were eligible for inclusion in the SEVARVIR cohort study (NCT05162508) if they presented the following inclusion criteria: age ≥18 years, SARS-CoV-2 infection confirmed by a positive reverse transcriptase-polymerase chain reaction (RT-PCR) in nasopharyngeal swab samples, admission in the ICU for acute respiratory failure (i.e., peripheral oxygen saturation (SpO$_2$) ≤90% and need for supplemental oxygen or any kind of ventilator support), patient or next of kin informed of study inclusion.

**Table 4 | Clinical and biological characteristics of the 148 patients infected with variant Omicron at the time of ICU admission according to the existence of underlying immunosuppression**

| | | No immunosuppression $N = 84$ | Immunosuppression $N = 64$ | *p*-value |
|---|---|---|---|---|
| Demographics and comorbidities | | | | |
| Sex, females | | 22 (26.19%) | 20 (31.25%) | 0.499 |
| Age, years | | 64.1 (±10.9) | 63.6 (±10.6) | 0.806 |
| Diabetes | | 27 (32.14%) | 24 (38.10%) | 0.453 |
| Obesity | | 39 (46.99%) | 12 (19.05%) | **<0.0001** |
| Chronic heart failure | | 11 (13.10%) | 4 (6.25%) | 0.172 |
| Hypertension | | 38 (45.24%) | 38 (59.38%) | 0.088 |
| Chronic respiratory failure | | 11 (13.10%) | 8 (12.50%) | 0.915 |
| Chronic renal failure | | 5 (6.02%) | 29 (46.03%) | **<0.0001** |
| Cirrhosis | | 1 (1.19%) | 2 (3.13%) | 0.408 |
| Number of comorbidities | | 1.50 (1;2) | 3 (2;4) | **<0.0001** |
| Clinical frailty scale | | 3 (2;4) | 3 (3;4) | **0.009** |
| SARS-CoV-2 infection and vaccination | | | | |
| Previous SARS-CoV-2 infection | | 5 (6.02%) | 1 (1.56%) | 0.175 |
| SARS-CoV-2 vaccination | | 34 (40.48%) | 55 (85.94%) | **<0.0001** |
| Number of doses among vaccinated | | 2.50 (2;3) | 3 (3;3) | **0.010** |
| Last dose - ICU admission[b], days | | 144 (39;210) | 175 (86;217) | 0.265 |
| SARS-CoV-2 serology at ICU admission | Unavailable | 38 (45.24%) | 23 (35.94%) | 0.516 |
| | Negative | 25 (29.76%) | 23 (35.94%) | |
| | Positive | 21 (25.00%) | 18 (28.13%) | |
| First symptoms - ICU admission, days | 7 (4.50;10) | 7.50 (4.50;12) | 0.228 | |
| SARS-CoV-2 RNA detection in nasopharyngeal swabs, Ct | | 25 (20;27) | 20 (18;24) | **0.003** |
| Patients severity upon ICU admission and biological features | | | | |
| WHO 10-point scale | | 6 (6;6) | 6 (6;6.5) | 0.531 |
| SAPS II score | | 32.50 (26;42) | 39 (31;49.5) | **0.005** |
| SOFA score | | 4 (2;6) | 5.50 (4;7) | **0.001** |
| $PaO_2/FiO_2$ ratio, mmHg | | 107 (77;147) | 114 (83;167) | 0.362 |
| Arterial lactate level, mM | | 1.65 (1.20;2.35) | 1.55 (0.95;2.05) | 0.096 |
| Blood leukocytes, G/L | | 8.00 (5.90;12.00) | 7.25 (3.70;13.15) | 0.305 |
| Blood lymphocytes, G/L | | 0.60 (0.40;0.90) | 0.30 (0.10;0.50) | **<0.0001** |
| Blood platelets, G/L | | 216.50 (162.50;286) | 161 (117;249) | **0.006** |
| Serum urea level, mM | | 7 (5;11) | 14 (9;23) | **<0.0001** |
| Serum creatinine level, µM | | 77 (59;101) | 133.50 (96;223.5) | **<0.0001** |
| Bacterial coinfection | | 13 (15.48%) | 8 (12.50%) | 0.607 |
| Pulmonary embolism | | 6 (7.23%) | 3 (4.76%) | 0.539 |
| Lung parenchyma involvement, % | | 50 (40;75) | 62 (37;75) | 0.487 |
| Oxygen/ventilatory support | Oxygen | 15 (17.86%) | 12 (18.75%) | 0.306 |
| | High flow oxygen | 43 (51.19%) | 33 (51.56%) | |
| | NIV/C-PAP | 11 (13.10%) | 3 (4.69%) | |
| | Invasive MV | 15 (17.86%) | 16 (25.00%) | |
| ECMO | | 1 (1.20%) | 1 (1.56%) | 0.853 |
| Vasopressor support | | 8 (10.00%) | 11 (17.74%) | 0.179 |

Results are *N* (%), means (±standard deviation) or medians (interquartile range, i.e., quartile 1; quartile 3).

*ICU* intensive care unit, *Ct* cycle threshold, *WHO* World Health Organization, *SOFA* Sequential Organ Failure Assessment, *SAPS II* Simplified Acute Physiology Score II, *NIV* non-invasive ventilation, *C-PAP* continuous-positive airway pressure, *MV* mechanical ventilation, *ECMO* extracorporeal mechanical ventilation.

[a]Includes HIV infection, long-term corticosteroid treatment, and other immunosuppressive treatments.

[b]Time lag between the last vaccination dose and ICU admission.

[c]Defined as <30 Binding Antibody Units (BAU)/mL.

Two-tailed *p*-values come from unadjusted comparisons using Chi-square or Fisher's exact tests for categorical variables, and *t*-tests or Mann–Whitney tests for continuous variables, as appropriate. No adjustment for multiple comparisons was performed; Bolded *p*-values are significant at the $p < 0.05$ level.

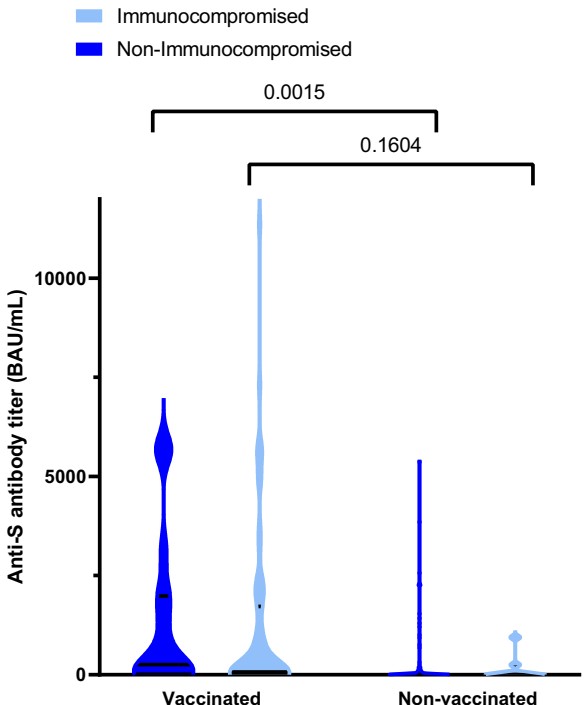

**Fig. 5 | Serum anti-spike (S) antibody titers in vaccinated and non-vaccinated, immunocompromised (light blue) and non-immunocompromised (deep blue) patients.** By two-way ANOVA, there was a significant effect of vaccination on anti-S titers ($p = 0.0023$), no significant effect of the immunocompromised status ($p = 0.6023$) and no significant interaction between both parameters (vaccination x immunocompromised status; $p = 0.9824$); Data distribution is represented using violin plots; Horizontal bars show median and quartiles 25 and 75; Displayed $p$-values have been obtained with the Sidak's *post-hoc* test.; *BAU* Binding Antibody Units, Source data are provided as a Source Data file.

Patients with SARS-CoV-2 infection but no acute respiratory failure or with a RT-PCR cycle threshold (Ct) value >32 in nasopharyngeal swabs were not included. The study was approved by the Comité de Protection des Personnes Sud-Méditerranée I (N° EudraCT/ID-RCB: 2021-A02914-37). Informed consent was obtained from all patients or their relatives.

Demographics, clinical and laboratory variables were recorded upon ICU admission and during ICU stay. Patients' frailty was assessed using the Clinical Frailty Scale[18]. The severity of the disease upon ICU admission was assessed using the World Health Organization (WHO) 10-point ordinal scale[19], the sequential organ failure assessment (SOFA[20]) score, and the simplified acute physiology score (SAPS) II score[21]. Acute respiratory distress syndrome (ARDS) was defined according to the Berlin definition[22]. The primary clinical endpoint of the study was day-28 mortality.

### SARS-CoV-2 variant determination

Full-length SARS-CoV-2 genomes from all included patients were sequenced by means of next-generation sequencing. Briefly, viral RNA was extracted from nasopharyngeal swabs in viral transport medium using NucliSENS® easyMAG kit on EMAG device (bioMérieux, Marcy-l'Étoile, France). Sequencing was performed with the Illumina® COVIDSeq Test (Illumina, San Diego, California), which uses 98-target multiplex amplifications along the full SARS-CoV-2 genome. The libraries were sequenced with NextSeq 500/550 High Output Kit v2.5 (75 Cycles) on a NextSeq 500 device (Illumina). The sequences were demultiplexed and assembled as full-length genomes by means of the

DRAGEN COVIDSeq Test Pipeline on a local DRAGEN server (Illumina). Lineages and clades were interpreted using Pangolin and NextClade, before being submitted to the GISAID international database (https://www.gisaid.org). Full-length Omicron genome sequence analysis yielding high coverage (>90%) were deposited in Genbank (accession numbers OP160034 to OP160218).

For mutational pattern analysis at the amino acid level, only high-quality sequences, i.e., sequences covering ≥90% of the viral genome, were considered. For variant comparison (Delta vs Omicron), mutation-specific RT-PCR was performed whenever the full-length genome was insufficiently covered by sequencing. For this, a multiplex mutation-specific RT-PCR kit (ID™ SARS CoV-2/VOC Revolution Pentaplex, IDSolutions, Grabels, France) was used to search for the presence of Spike amino acid mutations K417N, E484K, and L452R.

### Statistical analysis

Descriptive results are presented as means (±standard deviation [SD]) or medians (1st–3rd quartiles) for continuous variables, and as numbers with percentages for categorical variables. Two-tailed $p < 0.05$ were considered statistically significant. Unadjusted comparisons between patients infected with variants Delta and Omicron and between those infected with Omicron with or without immunosuppression were performed using Chi-square or Fisher's exact tests for categorical variables, and $t$-tests or Mann–Whitney tests for continuous variables, as appropriate. Adjusted analyses of the association between variants (i.e., Delta and Omicron) or Omicron subvariant lineages and 28-day mortality relied on multivariable logistic regression models, entering variables associated with a $p < 0.20$ in univariable analysis, then applying a stepwise backward approach by retaining only variables statistically significant at the $p < 0.05$ level and those previously shown to be important confounding factors, including vaccination status, gender and immunosuppression. Adjusted odds ratios (aOR) along with their 95% confidence intervals (CI) were computed. An exploratory evaluation of the associations between 28-day mortality and results from mutational pattern analysis at the amino acid level in Omicron-infected patients ($N = 97$ with available data) was performed by unadjusted logistic regression modeling. To limit the occurence of false positive findings, we analysed only those mutations outside Spike present or absent in at least 20 patients (i.e., $N = 11$) and we applied a correction of p-values for test multiplicity using the Benjamini–Hochberg procedure.

To illustrate differences in phenotypes of patients infected with variants Omicron or Delta, we performed an exploratory unsupervised clustering analysis using the Kohonen's self-organized map (SOM) methodology[23], allowing us to build 2-dimensional maps from multidimensional datasets. In a nutshell, each map is divided into districts in which patients are located by the SOM algorithm on the basis of their characteristics: patients with similar features are closely located on the maps, while patients with distinct profiles are farther from each other, hence allowing to identify key differences or similarities among them. The SOMs were obtained with the Numero package framework for the R statistical platform[24] after variables with missing information were imputed using the k-nearest neighbors (k-NN) approach and principal component analysis adapted for mixtures of qualitative and quantitative variables was applied (PCAMix)[25,26]. All measurements were taken from distinct samples. Analyses were performed using Stata V16.1 statistical software (StataCorp, College Station, TX, USA), and R 4.2.0 (R Foundation for Statistical Computing, Vienna, Austria).

### Reporting summary

Further information on research design is available in the Nature Research Reporting Summary linked to this article.

**Table 5 | Intensive care management and outcomes of the 148 patients with severe variant Omicron infection during their ICU stay according to the existence of underlying immunosuppression**

| | | No immunosuppression N = 84 | Immunosuppression N = 64 | *p*-value |
|---|---|---|---|---|
| Invasive MV | | 36 (42.86%) | 33 (51.56%) | 0.293 |
| Prone positioning | | 27 (33.75%) | 23 (41.07%) | 0.383 |
| MV duration, days | | 14.50 (6;26) | 10.50 (5;18) | 0.231 |
| ECMO support | | 6 (7.14%) | 3 (4.69%) | 0.536 |
| Duration of ECMO, days | | 8 (7.50;20.50) | 25 (25;25) | 0.468 |
| Vasopressor support | | 30 (36.59%) | 29 (45.31%) | 0.286 |
| Duration of vasopressors, days | | 3 (1;9) | 5 (1.50;15) | 0.309 |
| Renal Replacement Therapy | | 10 (11.90%) | 15 (23.44%) | 0.064 |
| Ventilator-acquired pneumonia (among IMV) | | 24 (68.57%) | 17 (53.13%) | 0.195 |
| Number of VAP episodes | Median (IQR) | 1 (0;2) | 1 (0;1) | 0.080 |
| | 0 | 11 (31.43%) | 15 (46.88%) | 0.223 |
| | 1 | 10 (28.57%) | 10 (31.25%) | |
| | 2 | 11 (31.43%) | 7 (21.88%) | |
| | 3 | 3 (8.57%) | 0 (0.00%) | |
| CAPA | | 2 (2.41%) | 5 (7.94%) | 0.122 |
| Dexamethasone | | 66 (85.71%) | 55 (88.71%) | 0.601 |
| Tocilizumab | | 31 (40.26%) | 18 (29.03%) | 0.168 |
| Monoclonal antibodies | | 9 (11.84%) | 19 (31.15%) | **0.005** |
| Casirivimab–Imdevimab | | 0 (0.00%) | 0 (0.00%) | – |
| Tixagevimab–Cilgavimab | | 8 (88.89%) | 16 (84.21%) | 0.741 |
| Sotrovimab | | 1 (11.11%) | 3 (15.79%) | 0.741 |
| Day-28 mortality | | 22 (26.19%) | 30 (46.88%) | **0.009** |
| Duration of ICU stay, days | | 10.50 (5;20) | 12 (6;20) | 0.509 |

Results are *N*(%), means (±standard deviation) or medians (interquartile range, i.e., quartile 1; quartile 3).

*MV* mechanical ventilation, *ECMO* extracorporeal mechanical ventilation, *VAP* ventilator-acquired pneumonia, *IMV* invasive mechanical ventilation, *CAPA* COVID-19-associated pulmonary aspergillosis.

ᵃVAP episodes were recorded per definition in patients under IMV since more than 48 h.

Two-tailed *p*-values come from unadjusted comparisons using Chi-square or Fisher'sexact tests for categorical variables, and *t*-tests or Mann–Whitney tests for continuous variables, as appropriate. No adjustment for multiple comparisons was performed; Bolded *p*-values are significant at the *p* < 0.05 level.

## Data availability

Source data are provided with this article. Due to the sensitive nature of the data, all the clinical datasets generated during and/or analyzed during the current study are available from the corresponding author on reasonable request (S.F.).

Full-length Omicron genomes were deposited in Genbank (accession numbers OP160034 to OP160218). Source data are provided with this paper.

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

## Acknowledgements
The authors would like to thank all study investigators, Dr Pierre-André Natella, Ms Nolwenn Bombenger for taking care of regulatory aspects, Ms Clélia Chambraud for taking care of data management, Mr Mohamed Ader for clinical data abstraction, the nurses and physicians who took care of the patients, the laboratory staff who took care of virological samples and the patients and their family for agreeing to participate in the study. The SEVARVIR study has been funded by the EMERGEN consortium - ANRS Maladies Infectieuses Emergentes (ANRS0153).

## Author contributions
N.D.P., E.A., J.M.P., and S.F., designed the study and obtained funding; E.A. performed statistical analyses; N.D.P., N.H., T.P., N.D.M., G.V., A.J., S.P., R.F., C.-E.L., J.M., D.R., S.Meireles, F.P., D.C., S.G., J.-F.T., A.K., L.-M.J., P.G., M.H., M.E., S.J., D.A., E.A., A.M.D., included the patient and were responsible for clinical data collection; E.G., A.C., L.M.-J., M.-L.C., A.G., S.B., S.Marot, D.D., F.R., A.H., S.B., C.H., S.F.-K., C.G.-S., A.P., C.R., and S.F. were responsible of the management of virological samples; J.-M.P., C.R., and S.F. were responsible of virological analyses; N.D.P., E.A., J.M.P., and S.F. wrote the first draft of the article; All authors revised and approved the article. The corresponding author attests that all listed authors meet authorship criteria and that no others meeting the criteria have been omitted. S.F. is the guarantor.

## Competing interests
S.F. has served as a speaker for GlaxoSmithKline, Abbvie, and Abbott Diagnostics. J.-M.P. has served as an advisor or speaker for Abbvie, Arbutus, Assembly Biosciences, Gilead and Merck. E.A. has received fees for lectures from Alexion, Sanofi, Gilead and Pfizer. His hospital has received research grant from Pfizer, MSD and Alexion. D.D. served as an advisor for Gilead-Sciences, ViiV Health care, Janssen-Cilag et MSD. F.P. served as an advisor for Gilead; he also received research grant from Alexion. C.-E.L. received lecture fees from MSD, Aerogen, Advanzpharma, and BioMérieux, outside the submitted work. J.-F.T. served as an advisor for pfizer, Gilead, BD, Gilead, Merck; he also received research grant from Thermofischer, merck, Pfizer, Biomerieux; lectures: pfizer, biomerieux BD, Merck, Shionoghi outside the submitted work. Other authors have no conflict of interest to disclose.

## Additional information

Article

Nicolas de Prost ®[1,2,3], Etienne Audureau[3,4,5], Nicholas Heming[6], Elyanne Gault[7], Tài Pham ®[2,8,9], Amal Chaghouri[10], Nina de Montmollin[11], Guillaume Voiriot[11], Laurence Morand-Joubert[12,13], Adrien Joseph[14], Marie-Laure Chaix[15,16], Sébastien Préau ®[17], Raphaël Favory[17], Aurélie Guigon[18], Charles-Edouard Luyt[19,20], Sonia Burrel ®[12,21], Julien Mayaux[22], Stéphane Marot[21], Damien Roux[23,24], Diane Descamps[25], Sylvie Meireles[26], Frédéric Pène[27], Flore Rozenberg[28], Damien Contou[29], Amandine Henry ®[30], Stéphane Gaudry[31], Ségolène Brichler[32], Jean-François Timsit[33], Antoine Kimmoun[34,35], Cédric Hartard[36], Louise-Marie Jandeaux[37,38], Samira Fafi-Kremer ®[39], Paul Gabarre[40], Malo Emery[41], Claudio Garcia-Sanchez[42], Sébastien Jochmans[43], Aurélia Pitsch[44], Djillali Annane[6], Elie Azoulay[14], Armand Mekontso Dessap[1,2,3], Christophe Rodriguez ®[3,45,46], Jean-Michel Pawlotsky ®[3,45,46] & Slim Fourati[3,45,46] ✉

[1]Médecine Intensive Réanimation, Hôpitaux Universitaires Henri Mondor, Assistance Publique—Hôpitaux de Paris (AP-HP), Créteil, France. [2]Groupe de Recherche Clinique CARMAS, Université Paris-Est-Créteil (UPEC), Créteil, France. [3]Université Paris-Est-Créteil (UPEC), Créteil, France. [4]Department of Public Health, Hôpitaux Universitaires Henri Mondor, Assistance Publique—Hôpitaux de Paris (AP-HP), Créteil, France. [5]IMRB INSERM U955, Team CEpiA, Créteil, France. [6]Médecine Intensive Réanimation, Hôpital Raymond Poincaré, Assistance Publique—Hôpitaux de Paris (AP-HP), Garches, France. [7]Laboratoire de Virologie, Hôpital Ambroise Paré, Assistance Publique—Hôpitaux de Paris (AP-HP), Boulogne, France. [8]Service de Médecine Intensive-Réanimation, Assistance Publique—Hôpitaux de Paris, Hôpital de Bicêtre, DMU 4 CORREVE Maladies du Cœur et des Vaisseaux, FHU Sepsis, Le Kremlin-Bicêtre, France. [9]Inserm U1018, Equipe d'Epidémiologie respiratoire intégrative, CESP, 94807 Villejuif, France. [10]Laboratoire de Virologie, Hôpital Paul Brousse, Assistance Publique—Hôpitaux de Paris, Villejuif, France. [11]Sorbonne Université, Centre de Recherche Saint-Antoine INSERM, Médecine Intensive Réanimation, Hôpital Tenon, Assistance Publique—Hôpitaux de Paris, Paris, France. [12]Sorbonne Université, INSERM, Institut Pierre Louis d'Epidémiologie et de Santé Publique, Paris, France. [13]Laboratoire de virologie, Hôpital Saint-Antoine, Assistance Publique—Hôpitaux de Paris, F-75012 Paris, France. [14]Médecine Intensive Réanimation, Hôpital Saint-Louis, Assistance Publique—Hôpitaux de Paris, Paris, France. [15]Université de Paris, Inserm HIPI, F-75010 Paris, France. [16]Laboratoire de Virologie, Hôpital Saint-Louis, Assistance Publique—Hôpitaux de Paris, F-75010 Paris, France. [17]U1167—RID-AGE Facteurs de Risque et Déterminants Moléculaires des Maladies Liées au Vieillissement, University Lille, Inserm, CHU Lille, Institut Pasteur de Lille, F-59000 Lille, France. [18]Service de virologie, CHU de Lille, F-59000 Lille, France. [19]Sorbonne Université, Assistance Publique—Hôpitaux de Paris, Hôpital Pitié–Salpêtrière, Médecine Intensive Réanimation, Paris, France. [20]INSERM UMRS_1166-iCAN, Institute of Cardiometabolism and Nutrition, Paris, France. [21]Département de Virologie, Hôpital Pitié–Salpêtrière, Assistance Publique-Hôpitaux de Paris (AP-HP), Paris, France. [22]Sorbonne Université, Assistance Publique–Hôpitaux de Paris, Hôpital Pitié–Salpêtrière, Médecine Intensive Réanimation, Paris, France. [23]Université de Paris, APHP, Hôpital Louis Mourier, DMU ESPRIT, Service de Médecine Intensive Réanimation, Colombes, France. [24]INSERM U1151, CNRS UMR 8253, Institut Necker-Enfants Malades (INEM), Department of Immunology, Infectiology and Hematology, Paris, France. [25]Université de Paris, IAME INSERM UMR 1137, Service de Virologie, Hôpital Bichat-Claude Bernard, Assistance Publique—Hôpitaux de Paris, Paris, France. [26]Service de Réanimation médico-chirurgicale, Assistance Publique–Hôpitaux de Paris, Hôpital Ambroise Paré, Boulogne, France. [27]Médecine Intensive Réanimation, Hôpital Cochin, Assistance Publique—Hôpitaux de Paris, Paris, France. [28]Laboratoire de Virologie, Hôpital Cochin, Assistance Publique—Hôpitaux de Paris, Paris, France. [29]Service de Réanimation, Hôpital Victor Dupouy, Argenteuil, France. [30]Service de Virologie, Hôpital Victor Dupouy, Argenteuil, France. [31]Service de Réanimation, Hôpital Avicenne, Assistance Publique—Hôpitaux de Paris, Bobigny, France. [32]Laboratoire de Virologie, Hôpital Avicenne, Assistance Publique—Hôpitaux de Paris, Bobigny, France. [33]Service de Médecine Intensive Réanimation, Hôpital Bichat, Assistance Publique—Hôpitaux de Paris, Paris, France. [34]Université de Lorraine, CHRU de Nancy, Médecine Intensive et Réanimation Brabois, Vandœuvre-lès-Nancy, France. [35]INSERM U942 and U1116, F-CRIN-INIC RCT, Vandœuvre-lès-Nancy, France. [36]Service de Virologie, CHRU de Nancy, Vandœuvre-lès-Nancy, France. [37]INSERM (French National Institute of Health and Medical Research), UMR 1260, Regenerative Nanomedicine (RNM), CRBS (Centre de Recherche en Biomédecine de Strasbourg), FMTS (Fédération de Médecine Translationnelle de Strasbourg), University of Strasbourg, Strasbourg, France. [38]Department of Intensive Care (Service de Médecine Intensive - Réanimation), Nouvel Hôpital Civil, Hôpital Universitaire de Strasbourg, Strasbourg, France. [39]Service de Virologie, Nouvel Hôpital Civil, Hôpital Universitaire de Strasbourg, Strasbourg, France. [40]Sorbonne Université, Assistance Publique—Hôpitaux de Paris, Hôpital Saint-Antoine, Médecine Intensive Réanimation, 75571 ParisCedex 12France. [41]Service de Réanimation, Hôpital Saint-Camille, Bry-sur-Marne, France. [42]Laboratoire de Biologie, Hôpital Saint-Camille, Bry-sur-Marne, France. [43]Service de Réanimation Polyvalente, Hôpital Marc Jacquet, Melun, France. [44]Laboratoire de Microbiologie, Hôpital Marc Jacquet, Melun, France. [45]Department of Virology, Hôpitaux Universitaires Henri Mondor, Assistance Publique—Hôpitaux de Paris, Créteil, France. [46]INSERM U955, Team « Viruses, Hepatology, Cancer », Créteil, France. ✉e-mail: slim.fourati@aphp.fr

