## [Peer Review File · Nature Communications]

Clinical phenotypes and outcomes associated with SARS-CoV-2 variant Omicron, sublineages BA.1, BA.1.1, BA.2 in critically ill French patients with COVID-19REVIEWERS' COMMENTS

Reviewer #1 (Remarks to the Author):

In this manuscript de Prost et al. evaluated within a multicenter prospective cohort study clinical phenotypes and outcomes of patients admitted to the ICU due to infection with SARS-CoV-2 variant Omicron, sublineages BA.1, BA.1.1, BA.2, or infection with variant Delta. 43.2% of patients infected with Omicron were immunocompromised (50% had received a solid organ transplant, 30% had a hematological malignancy), as evidenced by a poor humoral immune response to two or more doses of vaccine. Overall mortality was not different between patients infected with Omicron versus Delta. Mortality of immunocompromised patients was almost twice that of non-immunocompromised subjects. The authors found no difference between the different sublineages of Omicron and 28-day mortality.

The manuscript is well written, nicely presented, and of immediate interest to the scientific and clinical community. The below, rather minor suggestions could help improve the manuscript.

Major comments

1) The authors should introduce some more references in the introduction and discussion sections where they refer to previous findings, e.g. lines 145-147, "Compared to previous SARS-CoV-2 strains, ...structural and non-structural genes".

2) Figure 3, despite the nicely written figure legend, is difficult to read. What I understood from the figure, its legend and the text is that Omicron clustered in a similar region as did vaccination doses and immunosuppression, as well as partially hypertension, diabetes, chronic renal failure. Could the authors maybe elaborate a bit more on these?

Minor comments

a) Line 106, abstract, should read "... with rarer occurrence ...".

Reviewer #2 (Remarks to the Author):

This is a multicenter prospective study examining the role of several factors such as clinical conditions and variant in severity of infection. The examined patients admitted into 19/20 participating intensive care units in France between December 7, 2021 and May 1, 2022 which includes 2 very large covid waves. This is an important study, especially because it is prospective examining 28 day mortality and the role of patient characteristics as well as virus in severe covid outcomes.

According to the abstract 19/20 participating units were included. Why was 1 participating unit not included?

The authors only examine mutations outside of spike to limit the number of comparisons where the mutations were observed in at least 20 patients to limit comparisons; however it would be interesting to test some of the Spike mutations which have been associated with pathogenesis as well, such as P681R. Given that they have sequence data in these patients, this could be of interest to many readers.

Figure 4: shows p-value as $p=0.0015$ (and also in the text), but in the figure legend it is $p=0.0023$. Is $p=0.0015$ the unadjusted p-value? Also, the horizontal bars showing the median and IQRs are hard to see in the figure.

Could the authors give more guidance on how, for readers who are unfamiliar with self-organized maps, to read Figure 2.

Minor:

"keen" should be "kin": line 167

Line 321 seems to be redundant to the sentence following it and there is a period missing, this is probably a cut and paste error.

Table 1: should put if these are means (SD) or median (IQR) when each is used.

Perhaps there should be more context added in the comparison of Omicron versus delta in terms of outcomes. It should be noted that Delta was predominant in the first 5 weeks (8 weeks total until omicron fully took over) and accrued 111 admissions while Omicron was followed for 19 weeks and accrued 149 subjects, more than double the time followed under delta.

Reviewer #1 (Remarks to the Author):

In this manuscript de Prost et al. evaluated within a multicenter prospective cohort study clinical phenotypes and outcomes of patients admitted to the ICU due to infection with SARS-CoV-2 variant Omicron, sublineages BA.1, BA.1.1, BA.2, or infection with variant Delta. 43.2% of patients infected with Omicron were immunocompromised (50% had received a solid organ transplant, 30% had a hematological malignancy), as evidenced by a poor humoral immune response to two or more doses of vaccine. Overall mortality was not different between patients infected with Omicron versus Delta. Mortality of immunocompromised patients was almost twice that of non-immunocompromised subjects. The authors found no difference between the different sublineages of Omicron and 28-day mortality.

The manuscript is well written, nicely presented, and of immediate interest to the scientific and clinical community. The below, rather minor suggestions could help improve the manuscript.

R: We thank the reviewer for their very positive comments on our manuscript.

Major comments

1) The authors should introduce some more references in the introduction and discussion sections where they refer to previous findings, e.g. lines 145-147, "Compared to previous SARS-CoV-2 strains, ...structural and non-structural genes".

R: We agree with the reviewer's suggestion. We added four relevant references accordingly.

In the introduction section: " Compared to previous SARS-CoV-2 strains, ... additional mutations in other structural and non-structural genes" (doi: [10.1038/s41586-022-04411-y](https://doi.org/10.1038/s41586-022-04411-y); <https://doi.org/10.1038/s41392-022-01105-9>; references 6 and 7 of the revised manuscript).

In the discussion section: "Because BA.4 and BA.5 sublineages derive from BA.2 (<https://doi.org/10.1038/s41591-022-01911-2>) and these variants will become dominant in the coming weeks (doi: <https://doi.org/10.1136/bmj.o1969>)", references 16 and 17 of the revised manuscript.

2) Figure 3, despite the nicely written figure legend, is difficult to read. What I understood from the figure, its legend and the text is that Omicron clustered in a similar region as did vaccination doses and immunosuppression, as well as partially hypertension, diabetes, chronic renal failure. Could the authors maybe elaborate a bit more on these?

R: We have rephrased the figure legend and hope this will help better understand how to read the figure. The following sentences have been added: "*In the upper left area of the maps were located mostly Omicron-infected patients, as shown by the high prevalence rates (red colors) in this area of the Omicron map, whereas patients infected with variant Delta were distinctly located in the lower half area, as indicated by low Omicron prevalence rates (blue colors) in this area. Interpretation of the key characteristics associated with Omicron/Delta patients can be drawn from the observation of the other maps: e.g., Omicron patients from the upper left area also had higher immunosuppression rates and mean number of vaccination doses as indicated by the red colors in the corresponding maps.*"

Minor comments

a) Line 106, abstract, should read "... with rarer occurrence ...".

R: The reviewer is correct. The sentence was modified accordingly.

Reviewer #2 (Remarks to the Author):

This is a multicenter prospective study examining the role of several factors such as clinical conditions and variant in severity of infection. The examined patients admitted into 19/20 participating intensive care units in France between December 7, 2021 and May 1, 2022 which includes 2 very large covid waves. This is an important study, especially because it is prospective examining 28 day mortality and the role of patient characteristics as well as virus in severe covid outcomes.

R: We thank the reviewer for their very positive comments on our manuscript.

According to the abstract 19/20 participating units were included. Why was 1 participating unit not included?

R: There seem to be a misunderstanding here. The abstract used to state "...patients consecutively admitted for severe COVID-19 in 20 participating intensive care units in France...", consistent with the Methods section of the article. We confirm that there were 20 participating centres. However, because of formatting constraints (maximal number of words in the abstract=200), we had to remove this information from the abstract.

The authors only examine mutations outside of spike to limit the number of comparisons where the mutations were observed in at least 20 patients to limit comparisons; however it would be interesting to test some of the Spike mutations which have been associated with pathogenesis as well, such as P681R. Given that they have sequence data in these patients, this could be of interest to many readers.

R: We thank the reviewer for their relevant suggestion.

We further analyzed relevant spike mutations including P681R, as well as other mutations previously reported to correlate with worse clinical outcomes (Q57H in Orf3a; P25L in Orf3a ; S194L, R203K and G204R in N ; see ref*) in the Omicron full-length genomes. As now stated in the revised Results section, we found that none of these substitutions was associated with day-28 mortality.

In addition, the four following references have been added to the revised version of the manuscript, corresponding to references # 8 to 11:

- * Majumdar, P.; Niyogi, S. ORF3a Mutation Associated with Higher Mortality Rate in SARS-CoV-2 Infection. *Epidemiol Infect* 2020, 148, e262, doi:10.1017/S0950268820002599.
- * Nagy, Á.; Pongor, S.; Gyórfy, B. Different Mutations in SARS-CoV-2 Associate with Severe and Mild Outcome. *Int J Antimicrob Agents* 2021, 57, 106272, doi:10.1016/j.ijantimicag.2020.106272.
- * Pang, X.; Li, P.; Zhang, L.; Que, L.; Dong, M.; Xie, B.; Wang, Q.; Wei, Y.; Xie, X.; Li, L.; et al. Emerging Severe Acute Respiratory Syndrome Coronavirus 2 Mutation Hotspots Associated With Clinical Outcomes and Transmission. *Front Microbiol* 2021, 12, 753823, doi:10.3389/fmicb.2021.753823.
- * Saito et al. *Nature* 2022 doi: 10.1038/s41586-021-04266-9.

Figure 4: shows p-value as p=0.0015 (and also in the text), but in the figure legend it is p=0.0023. Is

p=0.0015 the unadjusted p-value? Also, the horizontal bars showing the median and IQRs are hard to see in the figure.

R: We do understand the point raised by the reviewer. In fact, these p values point to different statistical tests and provide different informations. The p values provided in the figure legend are the main results of the two-way ANOVA: effect of vaccination; effect of immunocompromised status; and interaction term. Because at least one of these p values has reached significance (ie, effect of vaccination: p=0.0023, as indicated in the figure legend), post-hoc tests (here, using the Sidak's post-hoc test) have been run to make direct comparisons between groups: the p-value shown on the figure indicates the results of post-hoc comparisons. In order to be more systematic, we have also provided the result of the comparison of vaccinated vs non-vaccinated patients in immunocompromised patients. We have also slightly modified the figure legend.

We tried to improve the visibility of horizontal bars, however, the narrow distribution of the data makes these hard to visualize in some groups.

Could the authors give more guidance on how, for readers who are unfamiliar with self-organized maps, to read Figure 2.

R: We believe the reviewer refers to Figure 3. As also suggested by reviewer 1, we have rephrased the figure legend and hope this will help better understand how to read the figure. The following sentences have been added: *"In the upper left area of the maps were located mostly Omicron-infected patients, as shown by the high prevalence rates (red colors) in this area of the Omicron map, whereas patients infected with variant Delta were distinctly located in the lower half area, as indicated by low Omicron prevalence rates (blue colors) in this area. Interpretation of the key characteristics associated with Omicron/Delta patients can be drawn from the observation of the other maps: e.g., Omicron patients from the upper left area also had higher immunosuppression rates and mean number of vaccination doses as indicated by the red colors in the corresponding maps."*

Minor:

"keen" should be "kin": line 167

R: The reviewer is correct and the typo has been corrected.

Line 321 seems to be redundant to the sentence following it and there is a period missing, this is probably a cut and paste error.

R: The reviewer is correct and the sentence has been modified accordingly. Table 1: should put if these are means (SD) or median (IQR) when each is used.

R: Continuous variables are indeed presented as indicated by the reviewer, depending on their distribution. The legend of all tables has been modified as follows: "Results are N (%), means (\pm standard deviation) or medians (interquartile range, i.e., quartile 1;quartile 3)."

Perhaps there should be more context added in the comparison of Omicron versus delta in terms of outcomes. It should be noted that Delta was predominant in the first 5 weeks (8 weeks total until omicron fully took over) and accrued 111 admissions while Omicron was followed for 19 weeks and accrued 149 subjects, more than double the time followed under delta.

R: We are not sure we understand the reviewer's comment regarding the need for more context on the comparison of Omicron versus Delta in terms of outcomes. The following sentences of the Introduction section provide the reader with updated references on this aspect: *"Although variant Omicron appears to cause a less severe disease than its predecessor variant Delta 1–4, with less frequent requirement for intensive care 1, a substantial number of Omicron-infected patients experienced severe COVID-19 with acute respiratory distress syndrome (ARDS) requiring intensive care support. A recent observational study has suggested that the mortality of critically ill patients infected with variant Omicron was not significantly different from that of patients infected with variant Delta 5"*.

Regarding the dynamics of inclusion of patients with Delta and Omicron variants we have slightly modified the following sentence in the first paragraph of the Results section: *"Variant Delta was predominant in December 2021 (i.e., during the first 5 week of the inclusion period), but was then rapidly replaced by variant Omicron"*.